# Structure-Based Screening and Optimization of PafA Inhibitors with Potent Anti-Tuberculosis Activity

**DOI:** 10.3390/ijms252313189

**Published:** 2024-12-08

**Authors:** Hewei Jiang, Jin Xie, Lei Wang, Hong Chen, Yunxiao Zheng, Xuening Wang, Shujuan Guo, Tao Wang, Jing Bi, Xuelian Zhang, Jianfeng Pei, Shengce Tao

**Affiliations:** 1Shanghai Center for Systems Biomedicine, Key Laboratory of Systems Biomedicine (Ministry of Education), Shanghai Jiao Tong University, Shanghai 200240, China; jianghewei@lglab.ac.cn (H.J.); remiliascarlet@sjtu.edu.cn (L.W.); miss_hongc@163.com (H.C.); xxmy07@163.com (Y.Z.); wxnxuening@163.com (X.W.); shjguo@sjtu.edu.cn (S.G.); 2Lingang Laboratory, Shanghai 200031, China; 3Center for Quantitative Biology, Academy for Advanced Interdisciplinary Studies, Peking University, Beijing 100871, China; xiejin1991@pku.edu.cn; 4Institute of Systems Biology, Shenzhen Bay Laboratory, Peking University Shenzhen Graduate School, Shenzhen 518055, China; taowang@szbl.ac.cn; 5State Key Laboratory of Genetic Engineering, School of Life Sciences, Fudan University, Shanghai 200438, China; 11210700007@fudan.edu.cn (J.B.); xuelianzhang@fudan.edu.cn (X.Z.)

**Keywords:** *M. tuberculosis*, pupylation, PafA, inhibitor

## Abstract

Tuberculosis (TB), caused by *Mycobacterium tuberculosis* (*Mtb*), remains a major global health challenge, primarily due to the increasing prevalence of drug resistance. Consequently, the development of drugs with novel modes of action (MOAs) is urgently required. In this study, we discovered and characterized two potent inhibitors, Pi-1-58 and Pi-2-26, targeting the prokaryotic ubiquitin-like protein (Pup) ligase PafA of *Mtb*. Pi-1-58 was identified through computer-aided drug screening. The binding mode of Pi-1-58 and PafA was investigated through molecular docking, followed by experimental validations. Based on the core structure of Pi-1-58, we developed a more potent inhibitor, Pi-2-26, through structure-guided drug design. Both Pi-1-58 and Pi-2-26 exhibited selective and specific inhibition of PafA according to biochemical and cell-based assays. Importantly, the inhibitors demonstrated significant inhibition on *Mtb* survival in the presence of nitric oxide, mimicking the in vivo nitrogen limited environment that *Mtb* encountered in macrophage. Our findings provide a comprehensive understanding of the structural and functional aspects of these PafA inhibitors and establish a solid foundation for the development of novel therapeutics against tuberculosis.

## 1. Introduction

Tuberculosis (TB), caused by *Mycobacterium tuberculosis* (*Mtb*), remains a global health threat, posing a significant burden on healthcare systems worldwide. The COVID-19 pandemic has not only exacerbated the challenges in TB control due to the diversion of healthcare resources but it has also been reported that SARS-CoV-2 infection could impair the host’s ability to control *Mtb*, resulting in increased susceptibility to TB infection or worsening of existing TB cases [1]. A more significant challenge in TB treatment is the emergence of drug-resistant strains, including multidrug-resistant (MDR) and extensively drug-resistant (XDR) TB. MDR-TB is characterized by resistance to at least two of the most critical first-line antibiotics, rifampicin and isoniazid, which form the cornerstone of tuberculosis treatment. For MDR-TB, the treatment success rate is approximately 57%, far lower than the 85% success rate observed for drug-susceptible TB [2]. Meanwhile, for XDR-TB, the treatment success rate drops to 39% [2]. In the past 50 years, only two new drugs, bedaquiline and delamanid, have been approved for treating MDR-TB [3,4]. However, these drugs face challenges such as adverse side effects and limited efficacy against certain MDR-TB strains [5,6]. Consequently, the development of drugs with novel modes of action (MOAs) is urgently required.

The pupylation pathway, a posttranslational modification system unique to actinobacteria, including *Mtb*, has recently emerged as a potential target for the development of new anti-TB drugs [7,8,9]. PafA, a ubiquitin-like protein ligase, plays a critical role in this pathway by catalyzing the covalent attachment of the small protein Pup to target proteins, marking them for proteasomal degradation [8]. Targeting PafA has become attractive for developing novel TB therapeutics due to its essentiality for *Mtb* survival and lack of homology to human proteins [7,10]. A recent study employed a high-throughput screening to identify PafA inhibitors, uncovering two effective compounds, ST1926 and bithionol, which efficiently suppressed the growth of an attenuated *Mtb* strain (H37Ra) at micromolar concentrations. Nonetheless, this screening approach falls short in terms of specificity and selectivity towards PafA inhibition [11].

In our previous research, we explored the potential of PafA as a drug target by identifying a critical residue, serine 119, that can be exploited for the development of PafA inhibitors [10]. Building upon this finding, we have utilized computer-aided drug design to develop potent specific PafA inhibitors, potentially leading to the development of novel therapeutic strategies for TB.

## 2. Results

### 2.1. PafA Inhibitor Identified by the Computer-Aided Drug Screening

Building upon our previous study that identified the S119 neighboring pocket as a potential site for PafA inhibitor screening [10], we first constructed a homology model of *Mycobacterium tuberculosis* (*Mtb*) PafA using the only available structure of *Corynebacterium glutamicum* (*Cglu*) PafA (PDB ID: 4BJR) as a template [12]. This template was chosen due to the high sequence similarity between *Cglu* PafA and *Mtb* PafA (53% identity, 92% similarity) (Appendix A). We also compared the *Mtb* PafA structure predicted by AlphaFold with our homology model, finding minimal overall differences between the two protein structures (RMSD = 0.6 Å) (Appendix A). Subsequently, a structure-based virtual screening approach was adopted to discover PafA inhibitors. After docking through all the three precision modes of Glide HTVS, SP, and XP, several candidate compounds from ChemDiv core database were selected as candidates for experimental testing. The ChemDiv core database (https://www.chemdiv.com/catalog/diversity-libraries/bemis-murcko-clustering-library/, accessed on 17 September 2018) comprises representative molecules selected to encompass the structural diversity of the entire ChemDiv database. In addition to examining the docking score and structural diversity, we also evaluated the lipophilicity (logP) and aqueous solubility (logS) of each compound, two important properties affecting activities. The logP and logS were evaluated using XLOGP3 and XLOGS, separately. Finally, 62 commercially available compounds were obtained for activity evaluation (Figure 1A). We refer to these inhibitors as PafA inhibitors (Pi).

To assess the inhibitory potential of these candidates on PafA enzymatic activity, we carried out in vitro assays using PafA-catalyzed PanB pupylation, a well-established assay for PafA pupylation activity [9]. Among the 62 potential candidates, Pi-1-58 strongly inhibited *Mtb* PafA pupylation of PanB (Appendix A). To confirm this result, pupylation assays were carried out with serially diluted Pi-1-58, and we found that the loss of PafA-mediated pupylation of PanB is indeed proportional to the concentration of Pi-1-58 (Figure 1B). Pi-1-58 inhibited PafA pupylation of PanB with a half-maximal inhibitory concentration (IC_50_) of 184.45 ± 27.65 nM (Figure 1C).

It is possible that the inhibition of Pi-1-58 on PafA activity is limited to only a few or a set of protein substrates, thus weakening the therapeutic potentiality of Pi-1-58. To rule out this possibility, we directly tested its effects using *E. coli* total proteins (cell lysates) as the substrate. It has been found that global pupylation modifications can be observed in *E. coli* total proteins upon the exogenous introduction of PafA and Pup [13]. We found that, in comparison to the DMSO control, *E. coli* total proteins treated with Pi-1-58 exhibited significant global pupylation reduction in a dose-dependent manner (Figure 1D). This finding indicates that the inhibition of PafA enzymatic activity by Pi-1-58 is general, not restricted to specific substrates. Overall, these findings indicate that Pi-1-58 is an effective inhibitor of the pupylase activity of *Mtb* PafA.

### 2.2. Structural Basis for the Inhibition of PafA-Catalyzed Pupylation by Pi-1-58

To unveil the molecular mechanism underlying the inhibition of PafA-catalyzed pupylation by Pi-1-58, we performed molecular docking using the *Mtb* PafA protein structure, derived from homology modeling based on the *Cglu* PafA (PDB ID:4BJR), and Pi-1-58 [12]. The results show that Pi-1-58 binds precisely to the S119-adjacent Pup binding pocket (Figure 2A), in line with our previous study identifying this region as an effective inhibitor binding site. We then analyzed the key amino acid residues within the binding pocket that potentially interact with Pi-1-58. Figure 2B presents the local cartoon representation of the PafA–Pi-1-58 complex, emphasizing residues V14, H61, and S119 that form hydrogen bonds with Pi-1-58. The binding pocket was found to be highly conserved between *Mtb* PafA and *Cglu* PafA (Appendix A). We tested the inhibitory effect of Pi-1-58 on *Cglu* PafA and observed a similar dose-dependent inhibition pattern, further validating our initial model used for inhibitor screening and reinforcing the conservation of the binding pocket (Appendix A). As shown in Figure 2C,D, Pi-1-58 was found to occupy the binding pocket well. Beneath the pocket lies a hydrophobic cavity that accommodates the cyclopentyl group of Pi-1-58, fitting precisely within the hydrophobic environment. This cavity is formed by a pocket defined by the residues V14, F109, and L355, into which the cyclohexane group is embedded. Upon analyzing the scaffold of Pi-1-58 bound to PafA, we observed a tight interaction between the central region of the inhibitor and the binding pocket of PafA. In contrast, the extremities of the molecule appeared to have larger spatial allowances. This finding suggests the possibility for further optimization by modifying the terminal regions to enhance binding affinity and specificity.

Taken together, our structural analysis provides insights into the molecular mechanism of Pi-1-58-mediated inhibition of PafA and supports the notion that the S119-adjacent Pup binding pocket represents a promising target for the development of specific and effective PafA inhibitors.

### 2.3. Structure-Based Development of Pi-1-58 Analogs

We employed two-dimensional and three-dimensional spatial similarity approaches to search for potential compounds from the whole ChemDiv database with ~1000,000 small molecules to improve potency and pharmacological properties. We identified 131 potential compounds and assessed their inhibitory potential against PafA-catalyzed PanB pupylation using in vitro enzymatic activity assays (Figure 3A, Appendix A). Among these candidates, Pi-2-26 demonstrated the most potent inhibitory activity, with a half-maximal inhibitory concentration (IC_50_) of 46.16 ± 7.96 nM (Figure 3B).

To evaluate the substrate selectivity of Pi-2-26, we also utilized the *E. coli* system and observed that Pi-2-26 exhibited global inhibitory activity (Figure 3C). Moreover, at equivalent concentrations, Pi-2-26 displayed a more potent global inhibitory effect than that of Pi-1-58 (Figure 3C). We further explored the potential mechanism of PafA inhibition by Pi-2-26 through molecular docking analysis. Our results revealed that, similar to Pi-1-58, Pi-2-26 bound to the pocket region where the C-terminus of Pup interacts with PafA (Figure 3D and Appendix A). Compared to Pi-1-58, the phenyl group of Pi-2-26 extends more deeply into the binding pocket, forming hydrogen bonds with D114 and S119, as well as a π–π interaction with H61. Additionally, the tail of Pi-2-26 penetrates further into the hydrophobic pocket, enhancing its interaction within the binding site (Figure 3D,E and Appendix A). The residue D114 formed two hydrogen bonds with two secondary amine groups in Pi-2-26, which is significantly different from the interactions with Pi-1-58. Additionally, the negatively charged residue D114 and the positively charged secondary amine group increased the stability of the PafA~Pi-2-26 complex (Appendix A). The enhanced inhibitory effect of Pi-2-26 compared to Pi-1-58 could be attributed to this tighter interaction binding mode.

To validate our docking predictions, we conducted co-immunoprecipitation (Co-IP) assays using GFP-Pup^E^ to enrich PafA protein. We observed a noticeable reduction in the amount of enriched PafA in the presence of either Pi-2-26 or Pi-1-58 compared to that of the DMSO control (Figure 3F). Moreover, the amount of PafA was even lower in the presence of Pi-2-26 as compared to that of Pi-1-58 (Figure 3F). These results suggest that the reduced PafA enrichment might be due to competitive binding of Pi-2-26 or Pi-1-58 to PafA. This observation provided further support for our molecular docking model.

### 2.4. Pi-1-58 and Pi-2-26 Inhibit Mtb Growth Under Nitric Oxide Stress

We next sought to investigate the inhibitory effects of Pi-1-58 and Pi-2-26 on the survival of *Mtb*, the causative agent of tuberculosis. A previous study has reported that the survival of *Mtb* is severely compromised in the presence of nitric oxide upon PafA knockout, whereas the impact on growth in normal culture medium is minimal [14]. Therefore, we initially examined the inhibitory effects of Pi-1-58 and Pi-2-26 on the attenuated strain H37Ra of *Mtb* under both nitric oxide stress and normal conditions.

We observed that in the presence of NO, both Pi-1-58 and Pi-2-26 demonstrated significant, dose-dependent inhibitory effects on *Mtb* survival compared to the DMSO control (Figure 4A). Furthermore, Pi-2-26 displayed a more potent inhibitory effect on bacterial survival than Pi-1-58 (Figure 4A), consistent with our in vitro biochemical data. In the absence of NO, the effects of Pi-1-58 and Pi-2-26 on *Mtb* survival were minimal, which is in line with the reported phenotypes of PafA knockout strains (Appendix A). This observation further corroborates the specificity of Pi-1-58 and Pi-2-26 targeting PafA. Moreover, we assessed the effects of Pi-1-58 and Pi-2-26 on the virulent *Mtb* strain H37Rv under NO-containing conditions and obtained similar results to those observed with H37Ra (Figure 4B).

In summary, our findings highlight the potential of Pi-1-58 and Pi-2-56 as effective inhibitors of *Mtb* survival, particularly under NO stress conditions, and provide further support for their specific targeting of PafA.

## 3. Discussion

In this study, we focused on identifying and optimizing inhibitors targeting PafA, a crucial enzyme in the pupylation pathway of *Mtb*. Utilizing computer-aided drug design, we identified Pi-1-58 as a potent inhibitor of PafA with an IC_50_ of 184.45 ± 27.65 nM. Further structural analysis and molecular docking revealed that Pi-1-58 binds specifically to the S119-adjacent Pup binding pocket of PafA. Building on these findings, we developed Pi-2-26, a more potent analog with an IC_50_ of 46.16 ± 7.96 nM. Both inhibitors demonstrated significant dose-dependent inhibition of *Mtb* growth under nitric oxide stress, validating their potential as novel therapeutic agents targeting the pupylation pathway in *Mtb*.

Since pupylation is primarily present in Nitrospira and Actinobacteria and is absent in most other bacteria, including the gut microbiota [10,15], targeting this pathway offers a highly specific approach to combat *Mtb*. Furthermore, PafA, a key enzyme in the pupylation pathway, shares no homology with ubiquitin ligases in eukaryotes [7,16,17], suggesting that drugs targeting PafA may have minimal side effects on human cells. These unique properties make PafA an attractive target for drug development. However, the extensive interaction interface between Pup and PafA poses a significant challenge for structure-based virtual screening. Our previous research pinpointed the critical S119 site in PafA for effective inhibitor targeting [10], enabling the development of focused computational drug screening strategies.

The discovery of Pi-1-58 via high-throughput computer-aided drug screening emphasizes the value of innovative strategies for finding novel inhibitors for challenging targets. Further molecular docking simulations and experimental validations offered insights into how Pi-1-58 binds to PafA. These findings laid the groundwork for optimizing the initial approach.

The discovery of Pi-2-26, a more potent inhibitor based on the core structure of Pi-1-58, showcases the potential of structure-guided drug design. Our study demonstrated that Pi-2-26 exhibits superior inhibitory activity both in vitro and in *Mtb*, which might be attributed to its more stable binding mode and tighter interaction with the key residues in the binding pocket. The consistency between the biochemical data and the results obtained from molecular docking further supports the validity of our structure-guided approach.

However, there are also limitations to our study. We have not yet conducted in vivo animal experiments to assess the efficacy and safety of our lead compounds. Animal studies are crucial for understanding pharmacokinetics, bioavailability, and potential toxicity, providing essential data before progressing to clinical trials. Furthermore, we have not evaluated the potential of these compounds in combination therapies with existing TB drugs. Combination therapies could enhance treatment efficacy, reduce the duration of treatment, and help prevent the development of resistance. Addressing these limitations in future research will be critical to fully realizing the potential of PafA inhibitors as novel TB therapeutics.

In conclusion, our study lays a solid foundation for the development of novel anti-tuberculosis therapeutics targeting PafA. Future research should focus on further optimization of Pi-2-26, exploring its pharmacokinetic properties, and evaluating its safety and efficacy in preclinical and clinical studies. The successful development of PafA inhibitors could provide a much-needed addition to the current arsenal of anti-tuberculosis drugs, particularly in the face of emerging drug-resistant strains.

## 4. Methods

### 4.1. Protein Cloning, Expression, and Purification

The protein sequences were downloaded from GenBank. The corresponding DNA sequences were codon-optimized and synthesized by Sangon Biotech (Shanghai, China). C-terminal His6-tagged PafA was cloned into pTrc99a. C-terminal His6-tagged PanB and N-terminal His6-GFP-tagged Pup^E^ were cloned into pET28a. All recombinant proteins were expressed in *E. coli* BL21 by growing recombinant *E. coli* BL21 cells in 1 L LB medium to an A600 of 0.6 at 37 °C. Protein expression was induced by the addition of 0.2 mM isopropyl-β-d-thiogalactoside (IPTG) before incubating cells overnight at 16 °C. Proteins were purified on Ni-NTA affinity columns and stored at −80 °C. Pup^E^ and N-terminal 5-carboxyfluorescein-cys-PupE were synthesized by GL Biochem, Shanghai, China.

### 4.2. Sequence Alignment and Homology Modeling

Since there is no available crystal structure of *Mtb* PafA, we examined the known structure of PafA from *Cglu* that exhibits high overall sequence similarity to *Mtb* PafA (53% identity, 92% similarity). Sequence alignment between *Mtb* PafA and *Cglu* PafA was performed with the ClustalW program. Considering that S119, a promising inhibitor binding site, is located in the Pup binding groove, we chose the Pup–PafA*_Cglu_* complex crystal structure (PDB ID:4BJR) as a template to build the homology model of *Mtb* PafA [12]. The three-dimensional structure of *Mtb* PafA protein was built by homology modeling using Modeller 9.17.

### 4.3. Structure-Based Virtual Screening

Protein structure was prepared using the “Protein Preparation Wizard” tool of the Schrödinger (version 11.5) suite, all water molecules were removed, the broken side chains were repaired, and missing hydrogen atoms were added. The OPLS3 force field was used to minimize energy and generate the most stable energy state.

The Receptor Grid Generation module of Glide of the Schrödinger (version 11.5) suite was used to generate a binding pocket for molecular docking, and a grid box of 30 Å × 30 Å × 30 Å was set up around the S119 site.

The ChemDiv core database, a collation of 200,000 small molecules, was selected as the screening database. Using the LigPrep mode of Glide, all compounds in the ChemDiv core database (https://www.chemdiv.com/catalog/diversity-libraries/bemis-murcko-clustering-library/, accessed on 17 September 2018) were preprocessed carefully. For each compound in the ChemDiv core database, the tautomers were generated at PH = 7.0 ± 2.0 and the different combinations of chiralities were also generated by using Epik tool of the Schrödinger (version 11.5), and applying the OPLS3 force field. Finally, the well-prepared database was used for a docking-based virtual screening process.

Molecular docking was performed using the Glide module. The docking parameters were all set to default. First, all molecules were screened using the Glide HTVS mode. Then, the top 10% of the HTVS outputs were screened using the SP docking mode. Finally, the top 30% of the molecules from SP docking were processed using the XP docking mode. The top 30% of the XP docking outputs were retained for further analysis.

### 4.4. Similarity Searching

Shape screening in the Schrödinger (version 11.5) software package was used for a 3D similarity search [18]. The docked conformation of PHD-1-58 in complex was used for shape screening, and the whole ChemDiv database with ~1000,000 small molecules was screened. The shape similarity indexes between each compound in the database and the reference compound were calculated. A total of 14,325 compounds with indexes between 0.8 and 0.99 were selected as candidates for the second-round selection.

### 4.5. IC_50_ Determination

To determine the potency of Pi-1-58 and Pi-2-26 against PafA, the compounds were tested at concentrations ranging from 0.005 to 20 μM in a 96-well plate format. The reaction mix contained 0.1 μM PafA, 1 μM PanB, and 1 μM N-terminal 5-carboxyfluorescein-cys-Pup^E^ in pupylation buffer (50 mM Tris-HCl, pH 7.5, 100 mM NaCl, 20 mM MgCl_2_) with 1 μL of each dilution of the compound or DMSO in a total volume of 99 μL. The reaction was initiated by the addition of ATP to a final concentration of 2 mM. Quantitative pupylation assays were performed according to a previously described procedure [19], except that N-terminal 5-carboxyfluorescein-cys-Pup^E^ was used instead of N-terminal 5-iodoacetamidofluorescein-cys-Pup^E^. The fluorescence intensity of each sample was measured with a Synergy 2 microplate reader (Biotek Instruments, California, United States) at 494 nm (excitation) and 522 nm (emission). The data points were collected in triplicate and the averaged value was used to generate concentration–response plots for Pi-1-58 and Pi-2-26. The IC_50_ value was obtained by nonlinear regression curve fitting of a four-parameter variable slope equation to the dose–response data using GraphPad Prism (version 10.3.1.509).

### 4.6. Inhibitor Profile of Pupylation Activity of PafA

All chemicals or inhibitors were purchased from ChemDiv. To determine the potency of chemicals or inhibitors against PafA, the compounds were tested at concentrations of 50 nM, 200 nM, and 500 nM in a 96-well plate format. The reaction mix contained 0.1 μM PafA, 1 μM PanB, and 1 μM N-terminal 5-carboxyfluorescein-cys-Pup^E^ in pupylation buffer (50 mM Tris-HCl, pH 7.5, 100 mM NaCl, 20 mM MgCl_2_) with 1 μL of each dilution of the compound or DMSO in a total volume of 99 μL. The reaction was initiated by the addition of ATP to a final concentration of 2 mM. Quantitative pupylation assays were performed according to a previously described procedure [19], except that N-terminal 5-carboxyfluorescein-cys-Pup^E^ was used instead of N-terminal 5-iodoacetamidofluorescein-cys-Pup^E^. The fluorescence intensity of each sample was measured with a Synergy 2 microplate reader (Biotek Instruments) at 494 nm (excitation) and 522 nm (emission). The initial round of computer-aided screening of PafA inhibitors and the subsequent round of structure-based similarity searching of Pi-1-58 inhibitors are detailed in Appendix A, respectively.

### 4.7. Pupylation Assays with PafA Inhibitors

PanB pupylation assay reactions were carried out in pupylation buffer containing PanB (5 μM), Pup^E^ (10 μM) and PafA (1 μM) at 30 °C for 2 h. For pupylation assays using lysate as the substrate, reactions included *E. coli* total proteins (10 μg), Pup^E^ (10 μM), and PafA (1 μM) and were incubated at 25 °C for 20 min with 5 mM ATP in pupylation buffer. In pupylation assays with PafA inhibitors, reactions were carried out as described above, except that PafA was preincubated in PafA inhibitors for 30 min at 25 °C. Samples were analyzed by SDS-PAGE, followed by Coomassie brilliant blue staining and Western blotting.

### 4.8. Co-Immunoprecipitation

PafA (1 μM), in the presence of different PafA inhibitors (50 μM) or DMSO, was incubated with GFP-tagged Pup^E^ (10 μM) or GFP (10 μM). The mixture was then incubated with anti-GFP immunomagnetic beads (Sangon Biotech, Shanghai, China). After incubation, the beads were washed to remove unbound proteins, and the bound proteins were eluted for further analysis. Input samples were analyzed by SDS-PAGE, followed by CBB staining. Enriched samples were analyzed by Western blotting with anti-PafA antibody (ABclonal, Wuhan, China) and anti-GFP antibody (ABclonal, Wuhan, China).

### 4.9. Mycobactericidal Activity

Log phase *Mtb* H37Ra or H37Rv strains were resuspended at OD 0.01 in acidic Sauton’s medium at pH 5.5 with or without 0.5 mM NaNO_2_ overnight. The pretreated strains were then dispensed into wells that contained PafA inhibitors at the indicated final concentrations in a 96-well plate. For H37Ra, after 14 days at 37 °C, CFUs were plated for enumeration and counted 3 weeks later. For H37Rv, after 7 days at 37 °C, spot tests of strains in 7H10 agar (BD) were carried out with photographic documentation taken after 3 weeks.

### 4.10. Abbreviations Table

A table containing all abbreviations used in this article can be found in Appendix A.

## Figures and Tables

**Figure 1 ijms-25-13189-f001:**
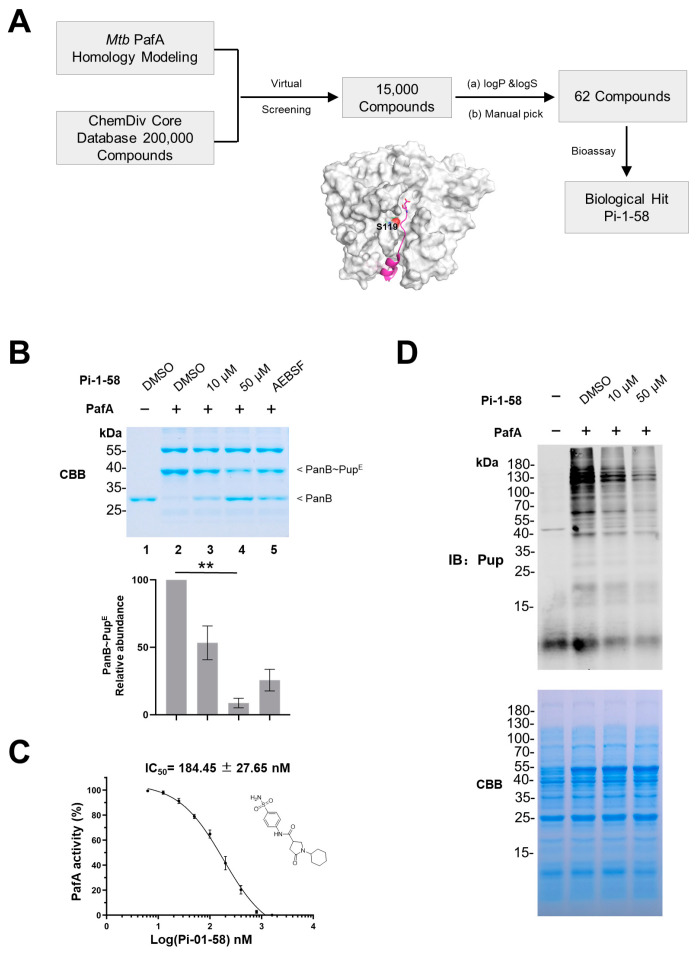
PafA inhibitor Pi-1-58 derived from the computer-aided inhibitor screening assay. (**A**) The workflow for computer-aided screening of PafA inhibitors. (**B**) PanB pupylation activity of PafA in the presence of various PafA inhibitors, using 2 mM ABESF as the positive control of PafA inhibitor. Samples were analyzed by SDS-PAGE, followed by Coomassie brilliant blue (CBB) staining. Quantitation of the pupylation level of PanB based on CBB staining is shown in the lower panel. Data are representative of three independent biological replicates (mean and s.e.m. of n = 3 samples), ** *p* < 0.01 (two-tailed unpaired *t*-test). (**C**) Pi-1-58 inhibits PanB pupylation activity of PafA with an IC_50_ = 184.45 ± 27.65 nM. The graph depicts percent activity relative to DMSO only control (mean ± SD). Chemical structure of Pi-1-58 is shown. (**D**) *E. coli* total protein pupylation activity of PafA in the presence of various PafA inhibitors. Samples were analyzed by SDS-PAGE, followed by CBB staining to serve as a loading control and Western blotting with an anti-Pup antibody.

**Figure 2 ijms-25-13189-f002:**
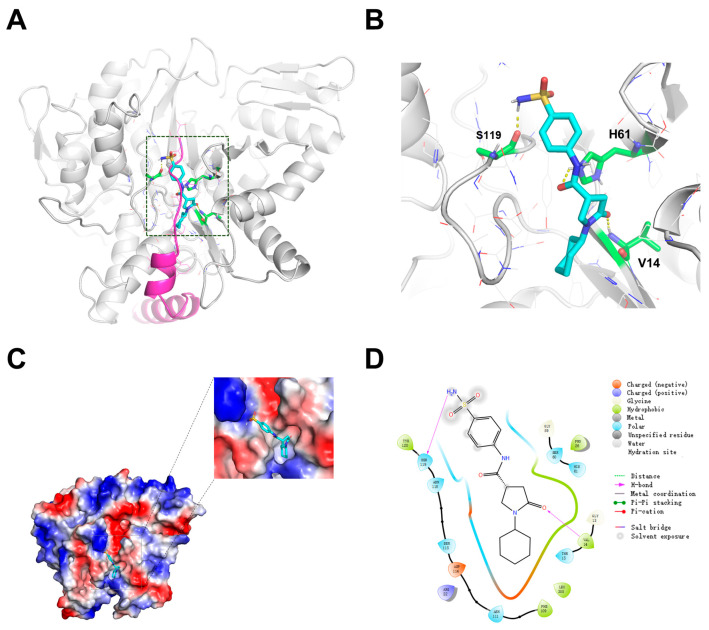
Structure–activity relationship analysis of Pi-1-58 on *Mtb* PafA. (**A**) Docking model of the PafA~Pi-1-58 (cyan stick) complex. Pup (pink) is shown as magenta cartoon. (**B**) Interaction details between Pi-1-58 (cyan stick) and PafA (white cartoon). Residues within 6 Å of Pi-1-58 are shown as white lines. Interacting residues are shown as green sticks and labeled. (**C**) Electrostatic surface potential map of PafA–Pi-1-58 complex. Red and blue areas mean negative and positive electrostatic potentials, respectively. (**D**) A 2D diagram of the interaction between Pi-1-58 and *Mtb* PafA.

**Figure 3 ijms-25-13189-f003:**
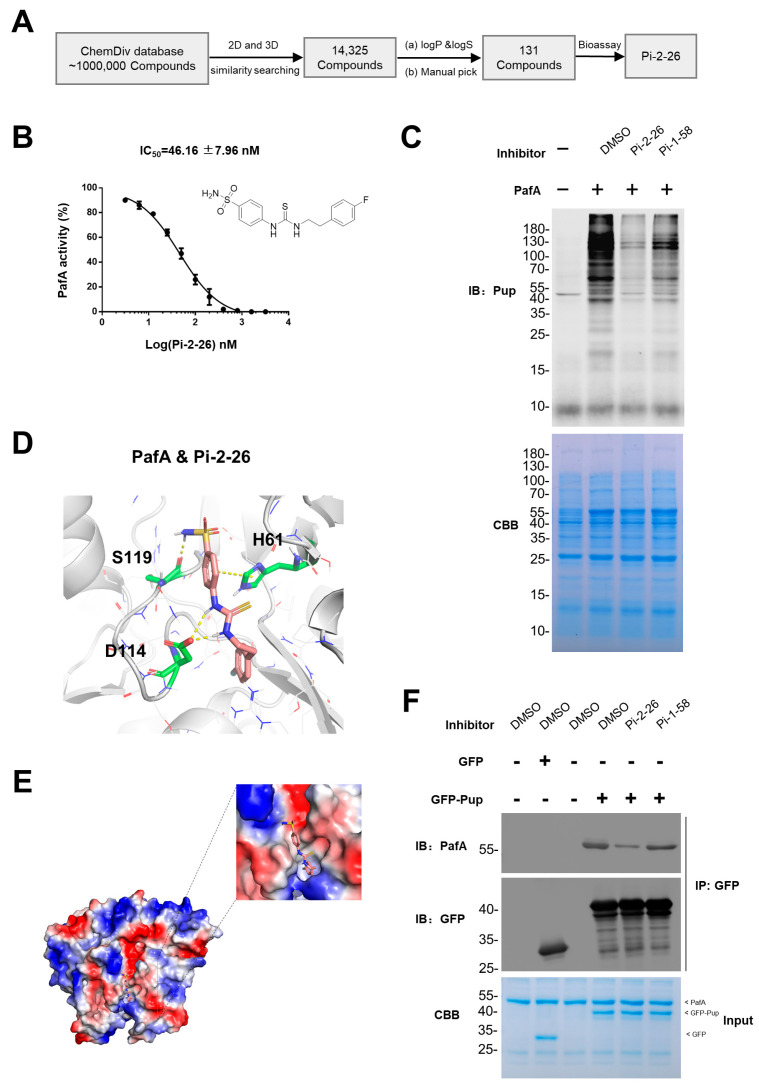
Structure-Based Development of Pi-1-58 Analogs. (**A**) The workflow for Structure-Based similarity searching of Pi-1-58. (**B**) Pi-2-26 inhibits PanB pupylation activity of PafA with an IC_50_ = 46.16 ± 7.96 nM. The graph depicts percent activity relative to DMSO only control (mean ± SD). Chemical structure of Pi-2-26 is shown. (**C**) *E. coli* total protein pupylation activity of PafA in the presence of various PafA inhibitors. Samples were analyzed by SDS-PAGE, followed by CBB staining to serve as a loading control and Western blotting with an anti-Pup antibody. (**D**) Interaction details between Pi-2-26 (salmon stick) and PafA (white cartoon). Residues within 6 Å of Pi-2-26 are shown as white lines. Interacting residues are shown as green sticks and labeled. (**E**) Electrostatic surface potential map of PafA-Pi-1-58 (cyan stick) complex. Red and blue areas mean negative and positive electrostatic potentials, respectively. (**F**) Co-immunoprecipitation verification of the Pi-binding site. PafA, in the presence of different PafA inhibitors, was incubated with GFP-tagged Pup or GFP. The mixture was then incubated with anti-GFP immunomagnetic beads. After incubation, the beads were washed to remove unbound proteins, and the bound proteins were eluted for further analysis. Input samples were analyzed by SDS-PAGE, followed by CBB staining. Enriched samples were analyzed by Western blotting with anti-PafA antibody and anti-GFP antibody.

**Figure 4 ijms-25-13189-f004:**
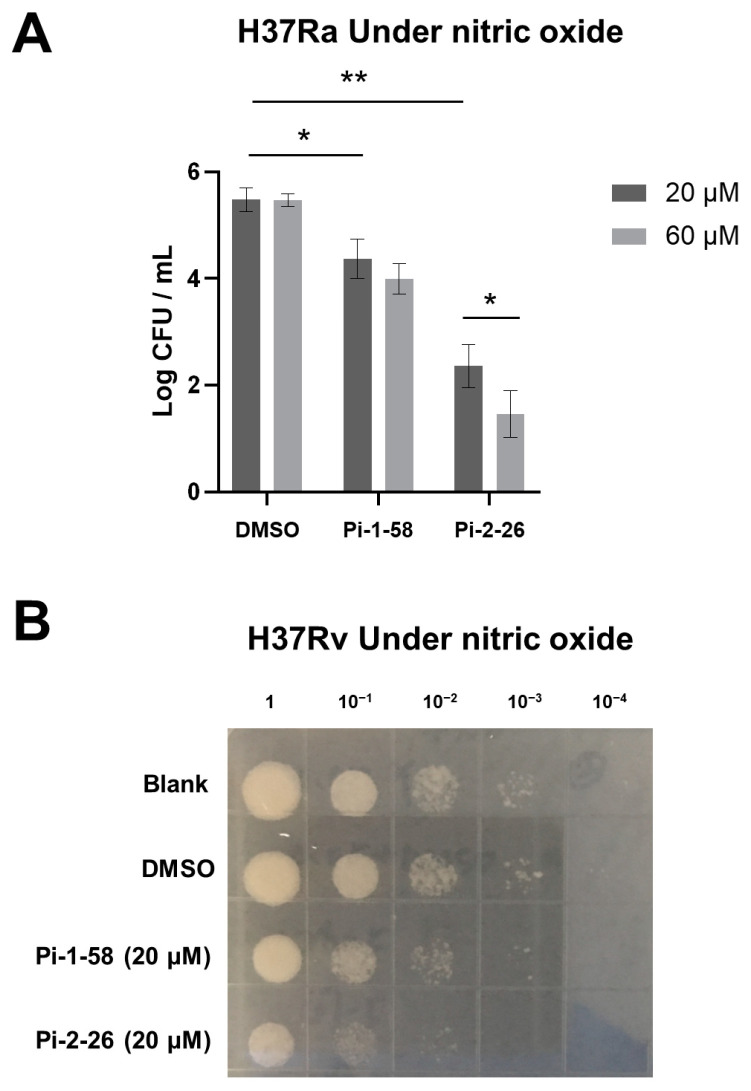
Nitric oxide stress assays on *Mtb* with PafA inhibitors. (**A**) Killing of *Mtb* H37Ra strains in Sauton’s medium with 0.5 mM NaNO_2_ at pH 5.5, which generates NO. *Mtb* H37Ra strains are treated with Pi or DMSO at the indicated concentrations. Data are representative of three independent biological replicates (mean and s.e.m. of n = 3 samples), * *p* < 0.05 and ** *p* < 0.01 (two-tailed unpaired *t*-test). (**B**) Spot test of *Mtb* H37Rv strains after 7 days of incubation in Sauton’s medium, which contained Pi or control and 0.5 mM NaNO_2_ at pH 5.5. This condition generates NO. Cell density of all cultures was normalized to OD_600_ of 0.5 prior to serial dilution.

## Data Availability

Additional data related to this paper may be requested from the authors.

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
