# Peer review of "Structure-Based Screening and Optimization of PafA Inhibitors with Potent Anti-Tuberculosis Activity"

_ijms, 2024, doi:10.3390/ijms252313189_

Round 1
Reviewer 1 Report
Comments and Suggestions for Authors
This article presents a very comprehensive work aimed at identifying new antitubercular compounds targeting the pupylation pathway, a post-translational modification characteristic of actinobacteria. To this end, a computer-aided drug design was performed, which resulted in compounds that are highly active against the PafA ligase enzyme, as confirmed by the in vitro assay. Furthermore, the two compounds were shown to inhibit Mtb growth under NO stress conditions.
Overall, the research is well conducted, and the results are very interesting and well discussed. I only have a few points:
The antimycobacterial activity was determined only at two concentrations, that are relatively high (20 and 60 uM). To have a clearer idea of the actual potency of the compounds, the MIC should be determined.
Moreover, have the authors checked the antimycobacterial activity also in standard conditions?
Author Response
Thank you very much for taking the time to review this manuscript. We greatly appreciate the insightful comments and thoughtful suggestions, which have helped enhance the clarity and depth of our work.
Comments 1: The antimycobacterial activity was determined only at two concentrations, that are relatively high (20 and 60 uM). To have a clearer idea of the actual potency of the compounds, the MIC should be determined.
Response 1: We sincerely appreciate the reviewer’s thoughtful suggestion regarding the determination of the MIC for our compounds. We understand that the MIC is a standard method for evaluating the potency of antimicrobial agents. However, in the case of our PafA inhibitors, we have some specific considerations that prevent the application of MIC testing in this context. Our compounds, even in the presence of nitric oxide, are not expected to exhibit bactericidal activity against Mycobacterium tuberculosis, as evidenced by the fact that PafA knockout strains do not result in bactericidal effects in vitro (PMID: 17277063). Given that PafA inhibition primarily affects bacterial metabolism and virulence rather than directly killing the bacteria, the traditional MIC assay would not accurately reflect the compound's effectiveness. Instead, we focused on assessing antimycobacterial activity through growth inhibition at two concentrations, which we believe provides more relevant information about the compound's potential therapeutic effect in this specific context.
We hope this clarification addresses your concern.
Comments 2: Moreover, have the authors checked the antimycobacterial activity also in standard conditions?
Response 2: We appreciate the reviewer’s insightful question regarding the evaluation of antimycobacterial activity under standard conditions. Indeed, we have assessed the effects of our inhibitors, Pi-1-58 and Pi-2-26, on Mycobacterium tuberculosis (Mtb) survival under standard conditions, specifically in the absence of nitric oxide (NO). As shown in Figure S4a (Page 4 of 13, line 171–173), the impact of these compounds on Mtb survival was minimal under these conditions. This result does not exhibit significant changes in Mtb viability in the absence of NO, reinforcing that PafA inhibition alone does not directly lead to bactericidal effects in standard culture conditions.
We hope this clarification resolves your concern.

Reviewer 2 Report
Comments and Suggestions for Authors
Mycobacterium tuberculosis remains one of the most notorious pathogens, posing a significant threat to humanity. As such, potent agents capable of controlling this pathogen are always welcomed. The authors identified and optimized a protein structure-based PafA inhibitor with high application value. However, I would like to offer some suggestions regarding certain details.
1. Explain which antibiotics multidrug-resistant Mycobacterium tuberculosis is resistant to.
2. Elaborate more carefully on the previous research mentioned at the end of the introduction. If this research has been published, please add a proper citation.
3. If no crystal structure is available, wouldn’t using AlphaFold-based structures be more accurate? Why was homology modeling chosen instead?
4. Figures 2C and 3E lack any text markings. Please add annotations and explanations to aid understanding.
5. Clearly state the source of the docking library used in the study.
Author Response
Thank you very much for taking the time to review this manuscript. We greatly appreciate the insightful comments and thoughtful suggestions, which have helped enhance the clarity and depth of our work.
Comments 1: 1. Explain which antibiotics multidrug-resistant Mycobacterium tuberculosis is resistant to.
Response 1: We appreciate the reviewer’s suggestion to provide additional information on the antibiotics to which multidrug-resistant (MDR) Mycobacterium tuberculosis is resistant. To address this, we have added a brief description of MDR-TB in the Introduction section. Specifically, we highlight that MDR-TB is resistant to at least the two most important first-line antibiotics, rifampin and isoniazid, which are central to the treatment of tuberculosis (Page 1 of 13, line 41-43). We believe this additional context will help clarify the significance of drug resistance in M. tuberculosis and the challenges it presents for treatment.
Comments 2: Elaborate more carefully on the previous research mentioned at the end of the introduction. If this research has been published, please add a proper citation.
Response 2: We appreciate the reviewer’s suggestion to elaborate on the previous research mentioned at the end of the introduction. We have revised the manuscript to clarify that the referenced screening approach did not specifically evaluate the specificity and selectivity towards PafA inhibition. (Page 2 of 13, 56-60, Nonetheless, this screening approach did not specifically evaluate the specificity and selectivity towards PafA inhibition.) This adjustment better reflects the limitations of the approach in the context of our study. Additionally, we have included the proper citation for this research (Reference #11) in the revised manuscript, as per the reviewer’s suggestion.
Comments 3: If no crystal structure is available, wouldn’t using AlphaFold-based structures be more accurate? Why was homology modeling chosen instead?
Response 3: We appreciate the reviewer’s insightful suggestion regarding the use of AlphaFold-based structures. However, our research began in 2017, at a time when AlphaFold had not yet been made publicly available. As a result, we relied on homology modeling, which was the best available method for predicting protein structures based on known homologous sequences at that time.
While we recognize that AlphaFold has significantly advanced the accuracy of structure prediction in recent years, the homology modeling approach was both widely used and effective for our study's goals at the time. We plan to revisit the structure predictions with more recent tools, including AlphaFold, as part of future work to further refine our findings.
Comments 4: Figures 2C and 3E lack any text markings. Please add annotations and explanations to aid understanding.
Response 4: We thank the reviewer for the valuable suggestion. In response, we have added the necessary annotations and explanations to Figures 2C and 3E for better clarity.
As shown in Figure 2C, the binding pocket of the PafA protein closely matches the shape of the small molecule. Below the pocket, there is a hydrophobic cavity where the cyclopentyl group of Pi-1-58 fits snugly (Page 3 of 13, line 119-122). In Figure 3E, the binding mode of Pi-02-26 is similar to that of Pi-1-58; however, Pi-02-26’s phenyl group extends more deeply into the binding pocket. Furthermore, the negative charge introduced by the F group of Pi-02-26 aligns well with the positive charge at the pocket’s end, and the two NH groups carry positive charges that complement the negative charges of the surrounding protein. This enhanced charge complementarity likely contributes to the tighter binding of Pi-02-26 to the PafA protein (Page 3 of 13, line 145-148).
Comments 5: Clearly state the source of the docking library used in the study.
Response 5: We appreciate the reviewer’s suggestion to clarify the source of the docking library used in our study. To address this, we have added a brief description of ChemDiv core data-base in Page 2 of 13, line 78-81. Specifically, for structure-based virtual screening, we selected the ChemDiv database (www.chemdiv.com), a publicly available resource containing approximately 1,000,000 chemically diverse small molecules. To optimize computational efficiency and reduce costs, we initially focused on a subset of this database. In the first round of virtual screening, we utilized the ChemDiv Core Database (https://www.chemdiv.com/catalog/diversity-libraries/bemis-murcko-clustering-library/), which consists of representative molecules chosen to cover the structural space of the entire ChemDiv database. In the second round, we performed similarity-based searches across the entire ChemDiv database to identify additional potential candidates based on the initial screening results.

Reviewer 3 Report
Comments and Suggestions for Authors
This article attempts to explore the development of novel PafA inhibitors with the help of in silico designs to develop new therapeutic strategies in tuberculosis. For a correct interpretation of the figures and graphs, including all abbreviations in the figure captions is recommended.
Author Response
Thank you very much for taking the time to review this manuscript. We greatly appreciate the insightful comments and thoughtful suggestions, which have helped enhance the clarity and depth of our work.
Comments 1:This article attempts to explore the development of novel PafA inhibitors with the help of in silico designs to develop new therapeutic strategies in tuberculosis. For a correct interpretation of the figures and graphs, including all abbreviations in the figure captions is recommended.
Response 1: Following your suggestion, we have listed all the abbreviations in a table for easier reference. This table can be found in Table S3.
